# Use of a Peer Equity Navigator Intervention to Increase Access to COVID-19 Vaccination Among African, Caribbean and Black Communities in Canada

**DOI:** 10.3390/ijerph22081195

**Published:** 2025-07-31

**Authors:** Josephine Etowa, Ilene Hyman, Ubabuko Unachukwu

**Affiliations:** 1School of Nursing, Faculty of Health Sciences, University of Ottawa, Ottawa, ON K1H 8M5, Canada; 2Dalla Lana School of Public Health, University of Toronto, Toronto, ON M5T 3M7, Canada; i.hyman@utoronto.ca; 3Canadians of African Descent Health Organization (CADHO), Ottawa, ON K2J 0R6, Canada; unachukwuubabuko@gmail.com

**Keywords:** peer education, health equity, health promotion, African American/black health, community intervention, critical race theory, outcome evaluation, program planning and evaluation, COVID-19, peer equity

## Abstract

African, Caribbean, and Black (ACB) communities face increased COVID-19 morbidity and mortality, coupled with significant barriers to vaccine acceptance and uptake. Addressing these challenges requires innovative, multifaceted strategies. Peer-led interventions, grounded in critical health literacy (CHL) and critical racial literacy (CRL), and integrating collaborative equity learning processes, can enhance community capacity, empowerment, and health outcomes, contributing to long-term health equity. This paper describes and presents the evaluative outcomes of a peer-led intervention aimed at enhancing COVID-19 vaccine confidence and acceptance. The Peer-Equity Navigator (PEN) intervention consisted of a specialized training curriculum grounded in CHL and CRL. Following training, PENs undertook a 5-month practicum in community or health settings, engaging in diverse outreach and educational activities to promote vaccine literacy in ACB communities. The evaluation utilized a modified Reach, Effectiveness, Adoption, Implementation, and Maintenance (RE-AIM) Framework, using quantitative and qualitative methods to collect data. Sources of data included tracking records with community feedback, and a PEN focus group, to assess program feasibility, outreach, and effectiveness. From 16 September 2022, to 28 January 2023, eight trained PENs conducted 56+ community events, reaching over 1500 community members. Both PENs and community members reported high engagement, endorsing peer-led, community-based approaches and increased vaccine literacy. The PEN approach proves feasible, acceptable, and effective in promoting positive health behaviors among ACB communities. This intervention has clear implications for health promotion practice, policy, and research in equity-deserving communities, including immigrants and refugees, who also face multiple and intersecting barriers to health information and care.

## 1. Background

COVID-19 disproportionately impacted African, Caribbean, and Black (ACB) populations in Canada, who comprised a higher share of cases, deaths, and hospitalizations compared to most of the population [1]. Canada lacks race-based data collection; thus, data on the extent of morbidity and mortality are drawn from a scoping review where higher rates of COVID-19 infections and lower rates of COVID-19 screening and vaccine uptake were reported among Black Canadians, largely attributed to pre-COVID-19 experiences of institutional and structural racism, health inequities, and a mistrust of healthcare professionals that further impeded access to healthcare. It was further noted that misinformation about COVID-19 exacerbated mental health issues among Black Canadians [2].

Addressing the complex health needs of equity-deserving populations, such as ACB communities, requires a multi-pronged approach and innovation. The aim of the paper is to describe the development of an innovative peer-led community engagement initiative, referred to as the peer-equity navigator (PEN) program, and share evaluative findings on its reach, acceptability, and effectiveness in reducing barriers to COVID-19 vaccine information and uptake in ACB populations in Ottawa, Canada.

According to Statistics Canada, ACB individuals constitute 3.5% of Canada’s total population and 15.6% of its visible minority population [3]. This diverse group includes immigrants and refugees, with intersecting identities such as gender, age, and religion influencing their health experiences [3]. Projections suggest that by 2036, ACB populations could represent between 5.0% and 5.6% of Canada’s total population, making them one of the fastest-growing racialized groups particularly in the Ottawa region [3].

Research indicates higher rates of COVID-19 vaccine hesitancy among ACB populations compared to the broader Canadian demographic. For instance, studies show significantly higher odds of non-vaccination against COVID-19 among Black populations compared to their White counterparts [4]. Multiple barriers contribute to this hesitancy, including systemic anti-Black racism, inadequate representation in healthcare leadership, limited awareness of available services, language barriers, and a lack of culturally competent healthcare providers [5,6].

Traditional top-down approaches to healthcare interventions, led primarily by researchers and decision-makers, may be less effective than community-driven initiatives. Community-based approaches, where community members and researchers collaborate to identify issues and develop solutions, are increasingly recognized as vital [7]. Scholars emphasize the importance of involving ACB communities in all aspects of health promotion and employing strengths-based strategies that leverage community institutions and assets, such as partnering with faith leaders, implementing peer-led interventions, and promoting economic empowerment [8,9,10,11].

The PEN program was developed by the Collaborative Critical Research for Equity and Transformation in Health (CO-CREATH) Lab at the University of Ottawa in collaboration with the Canadians of African Descent Health Organization (CADHO).

Previous research identified culture as a key determinant of health and help-seeking behavior in Canada and internationally [10,11,12,13,14]. One of the earliest frameworks to address culture, developed by Airhihenbuwa (1989) [13,15], placed culture at the core of the development, implementation, and evaluation of successful public health interventions. PEN-3, where the acronym PEN stands for Person, Extended Family, Neighborhood (Cultural Identity domain); Perceptions, Enablers, and Nurturers (relationship and expectation domain); Positive, Existential, and Negative (Cultural Empowerment domain), focusses on the need for culture-centered approaches to health that extends beyond assumptions surrounding individual responsibilities or capabilities to examine the role broader factors play in inhibiting and/or nurturing health behavior change [16].

Building on the PEN-3 framework, the CO-CREATH PEN program additionally draws on the literature demonstrating the positive impact of peer-led approaches, such as patient navigation, in addressing health inequalities. It is recognized that as early as 1990, patient navigation programs existed to improve patient outcomes such as cancer health disparities in minority populations [17]. While previous research conducted by our team and others used peer education models to engage ACB people in research activities especially in participants’ recruitment, the peer associates or educators often have limited critical racial and health literacy training. In contrast, this PEN program developed a comprehensive training program that addresses these topics in depth [18,19]. The community-based participatory research (CBPR) process also empowers the PENs to become change agents advocating in their community beyond the life of the project.

The primary goal of the PEN intervention was to enhance COVID-19 vaccine confidence and acceptance among ACB populations by addressing systemic barriers and fostering community-led solutions. The intervention involves a specialized, culturally consistent, training curriculum designed for peer navigators, combined with community-based vaccine promotion activities. The curriculum incorporates critical health literacy (CHL) and critical racial literacy (CRL) perspectives, addressing topics such as the historical context of ACB communities in Canada, social determinants of health (SDOH), health disparities, racism, and the socio-political dimensions of healthcare. The training format encourages learners to engage in a reflective process on racial equity, fostering collaborative and participatory learning. This approach aims to enhance community capacity, empowerment, and practice outcomes, thereby contributing to sustained improvements in community health and health equity [20].

Critical health literacy (CHL) involves the ability to access, comprehend, evaluate, and communicate health information to navigate various health contexts and support overall health throughout life [21]. This capability is shaped by socio-ecological factors such as poverty, low health literacy, cultural issues, and limited political influence [22,23]. In the context of COVID-19, vaccine uptake and related conditions such as HIV, CHL is increasingly essential for filtering pertinent information to guide personal health behaviors and promote well-being [24]. While most health literacy interventions focus on functional literacy, there is a notable scarcity of interventions incorporating interactive and critical health literacy concepts [25]. Reviews of community-based HIV and health literacy programs have shown positive outcomes, especially in terms of increased knowledge [26,27]. The most effective interventions involve high levels of community engagement, including active participation in evaluation, and emphasize empowerment, competence-building, and peer support [28].

Critical racial literacy (CRL) acknowledges that racism not only reflects personal biases but also systemic and institutional factors; thus, understanding, analyzing, and actively engaging with racial issues is critical to address the marginalization and social exclusion resulting from racism [29]. The aim of CRL is to foster an empowering environment where communities and individuals can confront daily prejudices and challenge the systemic structures that sustain racism. It plays a crucial role in promoting health equity, particularly among racialized groups such as ACB communities who have often been overlooked in research.

A key element of critical racial literacy (CRL) is the involvement of community members, often through peer models [20]. Peers, who are typically from the same community and share similar experiences and cultural backgrounds with the target population, are seen as more relatable and credible. This relatability helps in building effective communication and collaborative relationships [20]. Unlike experts with formal academic or professional training, peers bring valuable local insights, social connections, and cultural understanding. Their contributions enhance the effectiveness and scope of the research, while also fostering partnerships between the community and academic researchers by participating in various project activities [30].

The aim of this paper is to illustrate the impact of the PEN program on vaccine accessibility and community engagement.

## 2. Methods

### 2.1. Recruitment

PENs were recruited through a public job posting that was widely advertised within ACB networks and social media platforms. Selection criteria included the following: identification as ACB, desire to contribute to research and health equity in ACB communities, good communication skills, including the ability to clearly relay health information in a confident, friendly, and outgoing manner, facilitation skills, ability to actively listen with empathy and remain nonjudgmental, flexibility in working hours, and fluency in English and/or French. Eleven PENS were selected to be trained. Several were internationally trained health or social service professionals, seeking to gain experience in the Canadian healthcare system.

### 2.2. Training

The training curriculum was developed by CO-CREATH lab. Staff identified content and process areas for training including, social determinants of health in ACB communities, health issues in ACB communities, CHL, CRL, health promotion models and strategies, advocacy, and health policy. An environmental scan of evidence-based online and print resources was conducted to identify relevant teaching resources for the 12 training modules. For each module, brief 2-page and 5-page outlines with objectives, required and optional readings, discussion questions, and points for reflection were prepared, along with a PowerPoint presentation. Most training sessions were delivered online. The final session, Reflection and Knowledge Application, included role-playing exercises and case scenarios, was held in person.

Upon completion of the training, PEN trainees were assigned to a community or health organization in Ottawa for a 5-month practicum placement. The goal of the placement was for the PEN to gain familiarity with that organization’s mandate, activities, and approach to working with ACB communities. Building on their training, PENs were involved in a wide range of outreach and educational activities to increase vaccine and health literacy in ACB communities. Ten (out of the initial eleven recruited) PENs participated in the training program and eight out of ten completed the training and a preceptored practicum with local health facilities.

### 2.3. Evaluation

A modified version of the Reach, Effectiveness, Adoption, Implementation, and Maintenance (RE-AIM) framework was chosen [31]. This framework evaluates health interventions by assessing their Reach (feasibility in engaging in the target population), Adoption and Implementation (acceptability and quality of delivery), and Effectiveness (impact on outcomes) to determine how well an intervention can be implemented and sustained in real-world settings.

A logic model was developed to capture the outputs and outcomes of interest in the PEN intervention, including, number of community outreach sessions, workshops, and one-on-one interactions conducted, number of attendees at vaccine promotion events and workshops, increased vaccine knowledge, confidence, and literacy, and increased community engagement in health promotion. Indicator data was collected using tracking tools, participant surveys, and focus group discussions. Tracking sheets, developed by the research team, captured information on the number and type of community events hosted by the PENS, number of attendees per event, target population (if applicable), and positive, unintended, and negative impacts of the intervention. An excel spreadsheet was completed by each PEN following a community event, workshop or one-on-one interaction that included community feedback on the educational event. Changes in participants’ critical racial and health literacy were not directly assessed.

A focus group guide was developed to collect information from the PENS on their experiences in the program and with community engagement. During a 1.5 h online focus group, PENS were asked to describe the benefits of the program to the ACB communities in Ottawa and what contributed to these benefits. A total of 6 PENS participated in the focus group, which was recorded and later transcribed using a transcription software program, otter.ai.

Data from the tracking sheets was transferred to excel and collated for analysis. Braun and Clarke’s 6-step thematic analysis framework [32] was used to guide data analysis. This structured approach allows researchers to systematically identify patterns and meanings within qualitative data, providing a robust foundation for drawing conclusions and making interpretations. Findings related to the feasibility, acceptability, and immediate impact of the program were extracted.

## 3. Results

### 3.1. Reach

The eight PENS held more than 56 events between 16 September 2022 and 28 January 2023. This is likely an undercount because some events recorded as a single event included a series of sessions, and several events were co-hosted. The PENS were responsible for working with community partners to identify and host events in the community. Table 1 provides a description of the types of community events held by the PENS.

Although the exact figures could not be determined for some of the events (e.g., social media), findings suggest that over 1500 ACB people (range: 1438–1531+) were reached during community engagement.

### 3.2. Acceptability and Quality of Delivery

Most PENS reported high levels of community engagement. This was evident in the tracking sheets where PENs recorded the feedback from community members: ‘*Very high level of engagement with many questions asked’, ‘People seemed interested in the talk’, and ‘High level of engagement, participants were willing to talk and listen.’*

Community members expressed a great appreciation that the events were led by ACB community representatives, as evident in the following comments, ‘*Being educated by a fellow Black person’*, ‘*Blacks reaching out to Blacks*’, ‘*Identifying with a person of colour.*’

The acceptability of this educational format was evident in the following quotes made by participants in the PEN focus group.


*They felt she’s talking from the truth. She’s not talking as a professional. She’s not talking because her work is making her to say that. She’s talking because she’s trying to help me*
(FG participant)


*I think we had a very big and strong impact in the community because as peer navigators, we are not like scientists or decision makers in health. We’re more like shoulders to many of the community members so that they could rely on us to get the right information concerning their health*
(FG participant)

Several PENS noted the positive impact of having these events led by a Black community member. As one PEN explained,


*Many of them were excited to know that Blacks were with the Blacks doing something for ourselves, we’re not waiting on any other race or any other person to carry out this research for us.*
(FG participant)

Another PEN spoke of the access barriers that were reduced since PENS often spoke more than one language.


*There are some community members that don’t even speak in English, we speak in our different African languages, and it’s creates that connection.*
(FG participant)

This was also true for one-on-one community outreach approaches, as illustrated in the following quote:


*When we went individually, it was just you and the person you’re speaking with. I felt sometimes it’s a little bit better, especially if they don’t feel comfortable*
(FG participant)

### 3.3. Effectiveness

The effectiveness of the PEN approach was evident in the tracking notes recorded after several events: *‘After our discussion, they understood the importance of vaccines’, ‘We should hold such events periodically because of the high impact it has in the ACB community’, ‘It was a success. It was fun and interactive discussing with young people’*.

The benefits of the community engagement component were also captured by focus group participants who spoke of the effectiveness of using a peer-led, community-based approach to educate community members and build trust.


*It was interesting being with a group of friends and talking about our health, especially about COVID-19. Many of them were willing to admit that they were skeptical about receiving the COVID-19 vaccine. I was able to educate them about the effectiveness of the COVID-19 vaccine and correct any misinformation*
(FG participant)

However, PENs also suggested strategies to improve the impact of the PEN program. For example, program effectiveness could be increased if the length of the project was extended. This is evident in the following comment,


*I think, if the program was a little bit longer, they will have more impact. I know sometimes when we reached out, or I reached out to some people, they always [would] be like, is there going to be another event like this tomorrow, or next week. I felt like if we more time for this program, I think would have covered more ground*
(FG participant)

Several PENs agreed that the work of PENs may have more impact if it was conducted with trusted ACB organizations rather than mainstream health clinics.

## 4. Discussion

The PEN program was developed to address critical barriers to vaccine and health information and access faced by equity-deserving communities. Its unique features, critical to the program’s success, include the engagement and training of peers in critical health and racial literacy, the provision of Black-led culturally consistent educational initiatives, and partnerships with existing community and health service agencies.

The adage “knowledge is power” is particularly relevant for ACB communities. In a multicultural society like Canada, where systemic racism continues to shape health outcomes, these literacies are foundational to achieving health equity. CHL equips individuals with the ability to critically assess health information and understand the broader social determinants of health, such as poverty, housing, education, and access to care. CRL, on the other hand, enables individuals and institutions to recognize, analyze, and challenge the structural racism embedded in healthcare systems and policies. Together, CHL and CRL foster critical consciousness, an awareness necessary to understand and act upon the complex interplay between race, health, and social systems. This powerful combination empowers not only individuals but also community leaders, Peer Equity Navigators (PENs), and healthcare providers to co-create solutions that are just, inclusive, and sustainable.

A particularly distinctive and successful element of the PEN program is its unwavering commitment to authentic community engagement. From development through implementation, the program promotes shared ownership, trust, co-creation, and meaningful uptake among participants, ensuring that the intervention is not only impactful but also sustainable and rooted in the lived realities of the communities it serves.

Operationalizing Critical Health and Racial Literacy (CHRL) in a training curriculum and applying them effectively during the COVID-19 pandemic, requires intentional design, community engagement, and reflexive practice. We anchored the curriculum design in critical theories. For example, the integration of frameworks such as intersectionality, critical race theory, and the socio-ecological model help to contextualize health within systems of power and oppression. In addition, curriculum included the history of anti-Black racism in Canadian healthcare, and the social determinants of health specific to Black communities. Participatory and reflexive learning was fostered through storytelling and case studies that highlighted real-life scenarios of racialized health inequities. The use of community advisory groups throughout the project provided an opportunity for interactive evaluation to refine PEN training curriculum and ensure it remains relevant and empowering.

The RE-AIM framework was adopted to evaluate the outputs and outcomes of the evaluation. We were particularly interested in documenting the program’s feasibility in engaging the target population, acceptability, and quality of delivery. Evaluation findings suggested that this model is a feasible, acceptable, and effective intervention to improve vaccine confidence and acceptance among ACB people and could be potentially used to reach and engage ACB community members in other health promoting actions and behaviors. The trained PENS used their CHRL skills to address historical mistrust and acknowledge the racialized impact of the COVID-19 pandemic on Black communities. They combated misinformation by using their CHRL skills to identify and counter misinformation circulating in the community, especially on social media. The CHRL training helped them to frame COVID-19 not just as a biomedical issue but as a social justice issue, validating community concerns about vaccine access, testing, and treatment disparities. This approach increased the uptake of public health measures and strengthened trust between communities and health systems. PENs co-designed and led both virtual and in-person community outreach events that reflected community voices and needs. These PENs activities were effective because they were rooted in lived experience and mutual respect of the Black Canadian community.

Recent research highlights the potential of online community peer support interventions in addressing COVID-19 vaccine hesitancy. For instance, a randomized controlled trial demonstrated that peer-led online groups can serve as powerful platforms for disseminating accurate health information and aiding public health initiatives [33]. Building upon this research, ref. [34] investigated the efficacy of peer-to-peer approaches in mitigating vaccine misinformation and enhancing vaccine communication during pandemics. Their findings highlighted the importance of leveraging peer networks to combat vaccine hesitancy and address broader issues of public distrust in healthcare providers, government, and science [34]. Previous research shows that interventions that capitalize on the strengths of meaningful community engagement and focus on collaboration and partnering practices, contribute to empowerment, building community capacity, policy changes, and sustainable improvements in equity in health and healthcare [35].

The PEN program demonstrates significant potential for enhancing support services and addressing systemic barriers. Firstly, the program can serve as a valuable platform for providing training and resources to a diverse range of service providers, including healthcare professionals, social workers, and educators. This comprehensive approach ensures that these professionals are equipped with the necessary tools and knowledge to effectively support individuals and communities facing systemic challenges.

Furthermore, the PENS program can cultivate partnerships with local leaders, such as elected officials and community representatives. Collaborating with these stakeholders can raise awareness about the program and its services, thereby increasing its visibility and accessibility to those in need. Moreover, leveraging social media and other communication channels presents another avenue for improvement. By incorporating these platforms into the program’s outreach efforts, peer navigators can more effectively disseminate information about available services and establish connections with individuals who may not be familiar with the program. This proactive approach can help reach marginalized populations and ensure that support is accessible to all who require it.

## 5. Limitations

The evaluation of the PEN intervention has some limitations. For example, there is a lack of standardized measures for assessing vaccine literacy, trust and confidence. There were challenges in applying the evaluation framework in a community setting with limited resources to capture behavioral changes and long-term outcomes. Outcome data on intention to be vaccinated and/or vaccination status (as opposed to output) was difficult to collect from participants in community-based educational activities. The quality and completion of the self-reported event tracking was difficult to assess. While preliminary findings collected in the evaluation attest to the positive benefits of PENS educational activities, social desirability bias may affect the results; thus more rigorous evaluative designs are needed, including data on desired changes in critical health and racial literacy, and the long-term impacts of PENs on vaccine uptake and healthcare seeking. Future research is necessary to evaluate the impact of the PEN model on other health outcomes and in reducing ACB health inequities.

There are several lessons learned that could be applied to improve the efficiency and effectiveness of the PEN Model. First, longer terms of engagement are necessary to build trust with community members who may be initially resistant to vaccine promotion efforts [36]. Second, there is a need to improve the integration of PENs into community and health agencies. Third, while PENs are trained to move beyond knowledge transfer to be ‘change-makers in their communities’, they are not always able to address these within their organizations. However, key features of the intervention are relevant and applicable to reducing barriers to information and health services for other equity-deserving populations.

### 5.1. Implications for Research

Our findings highlight the need for studies to explore the adaptability and effectiveness of the PEN model in various settings and among different population groups, including immigrants and refugees. Expanding research efforts to assess the long-term impacts of peer-led interventions on vaccine uptake and broader health outcomes is crucial. Additionally, examining how peer navigators can be better integrated into existing communities and health agencies will provide insights into improving program implementation and overcoming integration barriers. Future research should also focus on how peer models can address systemic barriers to healthcare access and advocate for policy changes that promote health equity.

### 5.2. Implications for Practice

The PEN program highlights how peer navigators can significantly enhance health promotion efforts, particularly in reaching hard-to-reach communities, such as immigrants and refugees. Peer navigators, trained in critical health and racial literacy, effectively bridge gaps in health information and access, fostering increased vaccine confidence and uptake. Their deep understanding of community norms and barriers allows them to conduct targeted outreach and educational activities, addressing misinformation and facilitating access to services. To maximize their impact, health promotion strategies should integrate peer navigators into existing health frameworks, collaborate with local leaders, and utilize digital platforms for broader reach. This approach ensures that health promotion efforts are more inclusive and effective in addressing health disparities.

## 6. Conclusions

In conclusion, the PEN program to improve COVID-19 vaccine acceptance and uptake was found to be feasible, acceptable, and effective in reaching ACB community members in Ottawa, Canada; however, the work of PENs needs to be extended beyond the life of a given project. PENs are equipped with practical knowledge and skills to help their community, through the successful implementation of knowledge to action, dissemination strategies, and the identification of trusted sources of information. The critical awareness and transformation that occurs during critical health and racial literacy training is consistent with their role as community health advocates. Expansion of the length of the program and the improved integration of PENs within community and health agencies are suggested to improve the overall and long-term effectiveness of the program. Peer Equity Navigation is a beacon of empowerment, providing hard-to-reach community members with the compass to navigate through the complexities of healthcare, social injustice and advocacy for equitable change, one step at a time.

## Figures and Tables

**Table 1 ijerph-22-01195-t001:** Description of community events.

Description	Results
Lead agency	1/3 of events held with a host agency, ½ of events conducted with a community partner
Type of Host	Community Health Centers (N = 11), Public Health Units (N = 5), Medical Clinics (N = 4), AIDS Service Organizations (N = 2)
Target Audience	Francophones, seniors, Muslims, Christians, students/youth, women, internationally educated medical graduates
Format	½ of events conducted in person, ½ of events held online
Venues	Door-to-door, malls, large community events (N = 15), faith-based settings (N = 11), health clinics (N = 10), social media (N = 7+), friends/family (N = 7+), students/university settings (N = 3).

## Data Availability

Study data is available upon request.

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
