# Peer review of "Use of a Peer Equity Navigator Intervention to Increase Access to COVID-19 Vaccination Among African, Caribbean and Black Communities in Canada"

_ijerph, 2025, doi:10.3390/ijerph22081195_

Round 1
Reviewer 1 Report
Comments and Suggestions for Authors
Thank you for your paper, which analyzed using peer equity navigator (PEN) to increase COVID-19 vaccination among ACB. Although this study is a bit old, considering the COVID-19 pandemic was over sometime back, it provides useful findings that may apply to routine immunization program strengthening, particularly vaccine confidence and combating hesitancy. Below are some itemized suggestions to improve the paper:
- I suggest adding statistics on the burden of COVID-19 disease among the ACB population relative to other populations, i.e., white and other immigrant groups
- How does this mimic the acceptance and coverage of the routine immunization program among these groups?
- Were there similar interventions targeting the ACB that studies may have shown or which you’re aware of, and what does the PEN bring in terms of added value to other similar equity-driven interventions targeting the ACB
- Under methods, the PEN was not described but rather explanation and background information around CHL and CRL, which I suggest should be moved and summarized in the background section. The focus in the subsection PEN should rather concisely describe what the components of the PEN are and its features (lines 80 – 120)
- What was the PEN recruitment period, and if any specific targeting? Are these PEN volunteers, remunerated, etc? What type of contract do they hold, and what was the exit strategy after engagement?
- It was described that the study had an intervention component and an evaluation component with surveys and focus group discussions. Could you mention the specific study design?
- The results section can be better organized by presenting the descriptives in tables or charts.
- For the qualitative data, please describe how the data was coded and how themes were identified. The result for qualitative research should also be structured to flow based on the themes identified and properly report quotes from participants, as done in other manuscripts.
- Did you miss the results of the surveys conducted? Please present as well.
- The discussion was quite limited. I suggest it takes the themes and patterns observed from both the survey and the FGDs and discusses the findings. For instance, how was the application of the REAIM framework – strengths and weaknesses in this study?
- The ethical process was missed. I suggest including IRB clearance and how informed consent was obtained from participants for the FGDs
- I very much like the implication for research and practice sessions.
I recommend that the paper undergo thorough English proofreading.
Author Response
Reply to reviewers is provided in the attached word file.

Reviewer 2 Report
Comments and Suggestions for Authors
There are a couple of problems with this paper, which I very much hoped to like given our comparable experience in the US.
To begin, the authors do not reference work done on the problems related to minority health outside of Canada, nor well- known literature. In particular, nothing by Collins Airhihenbuwa is mentioned. [Here is a good reference to start: Iwelunmor J, Newsome V, Airhihenbuwa CO. Framing the impact of culture on health: a systematic review of the PEN-3 cultural model and its application in public health research and interventions. Ethn Health. 2014 Feb;19(1):20-46. doi: 10.1080/13557858.2013.857768. Epub 2013 Nov 22. PMID: 24266638; PMCID: PMC4589260.] Ironically, Airhihenbuwa created a "PEN-3" model decades ago, and calling the peer-equity navigator "PEN" should at least make mention that the authors are aware of the prior model.
More critically, this paper is written as if this is the first time anyone has engaged and training community members as navigators. There is extensive literature on this, beginning with the work of Harold Freeman out of Harlam, NYC, for cancer. https://www.jons-online.com/issues/2015/june-2015-vol-6-no-3/1327-the-birth-of-patient-navigation
The authors need to contextualize their work --how is it different from decades of efforts done in 'peer navigation' ? The unique aspects of the project are the emergency nature of COVID-19 engendering the tremendous lack of trust, and therefore, why 'critical health literacy' and 'critical racial literacy' were so important to the project. More could be explained about how these two perspectives were operationalized in the training curriculum and how they made a difference in presenting COVID-19 mitigation information to the community. Were they the 'secret sauce' and how do you know?
Where the paper really falls apart, however, is in the 'evaluation' of the program. The paper mentions voluntary end of activity surveys and tracking sheets. The tracking sheets indicate there were 56 events (in different venues and virtual) with perhaps 1500 people (this is thought to be an under count). Beginning around middle of page 5, there are numerous comments about the engagement and appreciation of the participants, but these are not put in italics, and it's difficult to tell where these comments come from. The PENs themselves--from the focus group held with them, or the PENs wrote these comments on the tracking sheets, or from the surveys. This whole section is interesting, but needs to be tightened up as to who said what when.
The authors had only 8 completed surveys, so as a representation of 1500, this number is inadequate and shouldn't be mentioned.
There is nothing said about how health literacy or racial literacy was improved. The authors say they couldn't measure improvements in vaccine literacy.
As a process model, they recruited 11, 10 participated in the training, 8 finished the program, (6 took part in one focus group). Not sure how feasible and replicable the process is. Of course, we agree that peer navigation is a beautiful framework and the problem is chiefly finding the money to pay navigators, particularly those that can cross disease topics and issues.
Bottom line, since you don't have the surveys and it's unclear what you can glean from the tracking sheets, can you do more with the focus group? Or can you interview the navigators again and see how they feel about the experience in retrospect. Would they want to do it again? Can they see ways to expand it? Can you go into the communities where events were held and interview the organizations that hosted the events. Do they have any ideas for sustainability? You need to give us more for this paper to add to what is already known about this problem and peer navigation as a solution. And, since this is late getting out about COVID, there have been several papers published about community-engaged efforts at this time. In the US, the National COVID Resiliency Network, led by the Morehouse School of Medicine, was most prominent.
Author Response
A reply to Reviewer 2 comments is provided in the attached word file.

Round 2
Reviewer 1 Report
Comments and Suggestions for Authors
Thank you for addressing the reviewer's comments. The paper provides a clear description of the PEN progamme among the ACBs, and will contribute to the equity and racial discourse around health. The methods and comprehensive discussions provide a clear pathway to replicate similar studies in similar contexts. Congratulations to the authors.
Author Response
There are no comments to respond to.
Reviewer 2 Report
Comments and Suggestions for Authors
Excellent job modifying the article to highlight the work. Two small things/typos. Line 115 vaccine should be lower case. Line 201 "inputted into survey monkeys" sounds really strange. Do you mean, data were input into Survey Monkey?
Author Response
Comment - Line 115 vaccine should be lower case.
Response - the correction was made
Comment - Line 201 "inputted into survey monkeys" sounds really strange. Do you mean, data were input into Survey Monkey?
Response - the problematic line was deleted